# Effect of the Transverse Functional Gradient of the Thin Interfacial Inclusion Material on the Stress Distribution of the Bimaterial under Longitudinal Shear

**DOI:** 10.3390/ma15238591

**Published:** 2022-12-02

**Authors:** Yosyf Piskozub, Liubov Piskozub, Heorhiy Sulym

**Affiliations:** 1Department of Applied Mathematics, Faculty of Computer Science and Telecommunications, Cracow University of Technology, Warszawska Str. 24, 31-155 Cracow, Poland; 2Department of Applied Mathematics and Physics, Ukrainian Academy of Printing, Pidgolosko Str. 19, 79020 Lviv, Ukraine; 3Department of Mechanics and Applied Computer Science, Faculty of Mechanical Engineering, Bialystok University of Technology, Wiejska 45C, 15-351 Bialystok, Poland

**Keywords:** functionally graded material, thin inclusion, composites, nonperfect contact, frictional heating, crack, stress intensity factor

## Abstract

The effect of a functional gradient in the cross-section material (FGM) of a thin ribbon-like interfacial deformable inclusion on the stress–strain state of a piecewise homogeneous linear–elastic matrix under longitudinal shear conditions is considered. Based on the equations of elasticity theory, a mathematical model of such an FGM inclusion is constructed. An analytic–numerical analysis of the stress fields for some typical cases of the continuous functional gradient dependence of the mechanical properties of the inclusion material is performed. It is proposed to apply the constructed solutions to select the functional gradient properties of the inclusion material to optimize the stress–strain state in its vicinity under the given stresses. The derived equations are suitable with minor modifications for the description of micro-, meso- and nanoscale inclusions. Moreover, the conclusions and calculation results are easily transferable to similar problems of thermal conductivity and thermoelasticity with possible frictional heat dissipation.

## 1. Introduction

Many structural materials contain numerous thin inhomogeneities in the form of inclusions of different origins [1,2,3,4,5,6,7]. Quite often, these inclusions are used as elements to reinforce the structural parts of machines and structures or as fillers in composite materials or coatings [8,9,10,11,12,13,14,15]. The use of nanocomposites with specific properties in engineering and technology has significantly shifted the interest from the study of objects at the macro level (100–10^−1^ m) and micro level (10^−3^–10^−6^ m) to the nano level (10^−9^ m) [16,17,18,19,20].

One of the typical examples of composite materials is the structure with thin ribbon inclusions. Structural elements made with the use of FGM have proven to be rather effective in practice [21,22,23]. In this way, it is possible to achieve a significant improvement in their mechanical, rheological, thermal, or other properties or the formation of protective thin layers [18,24,25,26,27].

The mathematical modeling of nanostructures requires the construction of more complex constitutional laws in comparison with the macro level [28,29]. Therefore, it is important to construct methods for studying the stress–strain state of such structures. To model a thin inclusion, there are mainly two basic approaches using analytical methods. The first one is based on the use of Eshelby’s analytical solution [21,30,31] for an ellipsoidal inclusion, in which a limiting transition with a decrease in one of the characteristic dimensions of the inclusion is performed. However, its application to thin interphase inclusions is impossible. The second one is based on the principle of the conjugation of continua of different measurability [32,33,34,35,36,37] and the method of jump functions [28,29,38,39,40,41]. According to this method, the inclusions are replaced by a certain surface (in the two-dimensional case, a line) of the discontinuity of the physical–mechanical fields, which describes the perturbing effect of a thin inclusion. Successful attempts were made in [17,33,42,43,44,45] to apply it to consider the influence of various physical and contact nonlinearities in the antiplane problem of elasticity theory for two compressed half-spaces with interfacial defects. The frictional slip with possible heat generation for contacting bodies [34,44,46,47,48,49] and the boundary element approach [50,51,52] were also considered here. Inhomogeneity of the mechanical properties of structural materials can be both designed for a specific purpose (FGM) and a consequence of technological processes of obtaining new materials and their processing (FSW, ball-burnishing process, etc.) [53,54]. Such factors cause additional complexity in the constitutive relations for the mathematical modeling of the behavior of such materials. However, the case of a thin inclusion of an inhomogeneous material has not been practically studied.

The process of improving the mathematical models of FGM [11,25,26,55,56,57,58] is complicated by the complex geometry of structural elements and the consideration of imperfections in the contact of their components. This is especially important to ensure their qualitative design, both in terms of mechanical strength [11,59,60,61,62,63] and in terms of the consideration of thermal, magnetic, and piezoelectric load factors [13,21,24,44,53,54,58,64,65].

The works [17,31,35,39] have been devoted to the consideration of surface energy and stresses in nanocomposites. Consideration of the heterogeneity of inclusions’ properties at the micro and nano scale is particularly important because the heterogeneity density of matter (discrepancy variation in mechanical and other properties) of matter bodies with a decrease in their scale usually increases, and the impact of this heterogeneity increases even further.

This publication aims to develop the method of jump functions and to construct a convenient structured and highly versatile approach to study the longitudinal displacement and thermal heating of bodies with thin inclusions of arbitrary physical nature, including those made from FGM.

## 2. Formulation of the Problem

Consider an unbounded isotropic structure consisting of two half-spaces with elastic moduli $G_{k}\;(k=1,2)$, which is subjected to an external longitudinal shear load determined by uniformly distributed infinity stresses $\sigma_{yz}^{\infty }$ and $\sigma_{yz}^{\infty }$, concentrated forces of intensity $Q_{k}$ and screw dislocations with vector Burgers component $b_{k}$ at points $\varsigma_{*k}\in S_{k}$(k=1, 2). The stresses must satisfy the conditions $\sigma_{xz2}^{\infty }G_{1}=\sigma_{xz1}^{\infty }G_{2}$ at infinity to ensure the straightness of the interface.

Let us investigate the stress–strain state (SSS) of the body section with a plane xOy perpendicular to the direction Oz of its longitudinal displacement (external problem). The plane sections of half-spaces perpendicular to this axis form two half-planes $S_{k}\;(k=1,2)$ and the abscissa axis corresponds to the interface L~x between them (Figure 1). At the interface of half-spaces (plane *xOz*), there is a tunnel section $L^{\prime }=[-a;\;a]$ in the direction of the shear axis *z*, in which a certain object of general thickness 2h (h≪a) is inserted.

According to the paradigm of the method of jump functions [36], the presence of a thin inclusion in the bulk is modeled by jumps in the components of stress and displacement vectors in [38,40,41]:(1)$$\begin{array}{l} {\left[\sigma_{yz}\right]}_{h}≅\sigma_{yz}^{-}-\sigma_{yz}^{+}=f_{3}^{}\left(x\right),\;x\in {L^{\prime }}_{}^{}\\ {\left[\frac{\partial w}{\partial x}\right]}_{h}≅\frac{\partial w^{-}}{\partial x}-\frac{\partial w^{+}}{\partial x}={\left[\frac{\sigma_{xz}}{G}\right]}_{h}\equiv \frac{\sigma_{xz}^{-}}{G_{1}}-\frac{\sigma_{xz}^{+}}{G_{2}}=f_{6}^{}\left(x\right); \end{array}$$
(2)$$f_{3}^{}\left(x\right)=f_{6}^{}\left(x\right)=0,\;\mathrm{if}\;x\notin L^{\prime }.$$

It is hereinafter marked ${\left[•\right]}_{h}=•\left(x,-h\right)-•\left(x,+h\right)$, $\;{\langle •\rangle }_{h}=•\left(x,-h\right)+•\left(x,+h\right)$; superscripts “+” and “−” correspond to the boundary values of the functions on the upper and lower banks of the line L.

The mathematical model of a thin inclusion is given as complicated conditions of imperfect contact between opposite matrix surfaces adjacent to the inclusion (internal problem) [28,38,39,40,65]. The general model of a thin, physically nonlinear inclusion is presented in [28,29,38], where the methods of modeling thin objects involve the integration of the defining relations describing the physical and mechanical properties of the material of the inclusion, with the subsequent consideration of the smallness of one of the linear dimensions of the inclusion.

Let us consider a similar model for a thin inclusion, assuming that the mechanical properties of the inclusion material are coordinate-dependent. This will allow us to model the inclusions of a functionally graded material:(3)$$\left\{ \begin{array}{l} G_{x}^{in}(x){\langle \frac{\partial w^{in}}{\partial x}\rangle }_{h}(x)-2\sigma_{xz}^{in}(-a)-\;\frac{1}{h}\int_{-a}^{x}{\left[\sigma_{yz}^{in}\right]}_{h}(\xi )d\xi =0,\\ G_{y}^{in}(x){\left[w^{in}\right]}_{h}(x)+h{\langle \sigma_{yz}^{in}\rangle }_{h}(x)=0. \end{array}\right.$$

Here, $G_{x}^{in}(x)$, $G_{y}^{in}(x)$ are the variable shear moduli of the inclusion’s material. As a special case, considering their values to be constant, we obtain Hooke’s law. The upper index “*in*” denotes the terms describing the inclusion material’s SSS components.

Contact between matrix components and the inclusion at $L^{\prime }$ and between the bimaterial structure components along a line $L\backslash L^{\prime }$ is supposed to be mechanically perfect,
(4)$$w(x,\pm h)=w^{in}(x,\pm h),\;\sigma_{yz}^{in}(x,\pm h)=\sigma_{yzk}(x,\pm h),\;x\in L^{\prime },\;\begin{array}{l} w(x,+0)=w(x,-0),\;\\ \sigma_{yz2}^{}(x,+0)=\sigma_{yz1}(x,-0),\;x\in L\backslash L^{\prime }, \end{array}$$
or frictional in some areas $x\in \;L_{f}\subset L^{\prime }$, as was considered in works [41,46,47,48],
(5)$$\sigma_{yz}^{in}(x,\pm h)=\sigma_{yzk}^{}(x,\pm h)=-\mathrm{sgn}{\left[w^{in}\right]}_{h}\tau_{yz}^{\max}.$$

Here, $\tau_{yzK}^{\max}$ is the limit value of shear stresses, at which the slippage begins. In this case, however, additional iterative methods should be applied to determine the area of the slip zones depending on the specific types of external loading of the composite [41].

## 3. Materials and Methods

Expressions for the components of the stress tensor and the derivatives of displacements on the line L of the infinite plane $S=S_{1}\;\cup S_{2}$, as well as inside the latter, can be obtained by applying the results of [37] to the solution of the external problem
(6)$$\begin{array}{l} \sigma_{yz}\left(z\right)+i\sigma_{xz}\left(z\right)=\sigma_{yz}^{0}\left(z\right)+i\sigma_{xz}^{0}\left(z\right)+\\ +ip_{k}g_{3}^{}\left(z\right)-\;Cg_{6}^{}\left(z\right)\;\left(z\in S_{k};\;r=3,\;6;\;k=1,\;2\right); \end{array}$$
(7)$$\begin{array}{l} \sigma_{yzk}^{\pm }\left(x\right)=\mp p_{k}f_{3}^{}\left(x\right)\;-Cg_{6}^{}\left(x\right)+\sigma_{yz}^{0\pm }\left(x\right),\;\\ \sigma_{xzk}^{\pm }\left(x\right)=\mp Cf_{6}^{}\left(x\right)+p_{k}g_{3}^{}\left(x\right)+\sigma_{xz}^{0\pm }\left(x\right),\; \end{array}$$
where the notation [28,41] is introduced:(8)$$\begin{array}{l} g_{r}^{}\left(z\right)\equiv \frac{1}{\pi }\int_{{L^{\prime }}_{}}\frac{f_{r}^{}\left(x\right)dx}{x-z}\;,\;s_{r}^{}\left(x\right)\equiv \;\int_{-a}^{x}f_{r}^{}\left(x\right)dx,\\ p=\frac{1}{G_{1}+G_{2}},\;p_{k}=G_{k}p,\;C=G_{1}G_{2}p. \end{array}$$

The superscript “+” corresponds to *k =* 2 and “–” corresponds to *k =* 1. Values marked with superscript “0” correspond to values in a continuous medium without inclusions under the same external load (homogeneous solution) [28,41].

Using (7), (8) and boundary conditions (4), it is easy to obtain from model (3) a system of singular integral equations:(9)$$\begin{array}{l} \{\begin{array}{c} (p_{2}-p_{1})f_{6}\left(x\right)+2pg_{3}\left(x\right)-\frac{s_{3}^{}\left(x\right)}{hG_{x}^{in}(x)}\;=F_{3}\left(x\right),\\ (p_{2}-p_{1})f_{3}\left(x\right)+2Cg_{6}\left(x\right)-\frac{G_{y}^{in}(x)s_{6}^{}\left(x\right)}{h}\;=F_{6}\left(x\right), \end{array}\\ F_{3}\left(x\right)=\frac{2}{G_{x}^{in}}\sigma_{xz}^{in}(-a)-\left(\sigma_{xz2}^{0}(x)/G_{2}+\sigma_{xz1}^{0}(x)/G_{1}\right),\\ F_{6}\left(x\right)=\langle \sigma_{yz}^{0}\rangle (x)-G_{y}^{in}\langle \frac{\sigma_{yzk}^{0}(x)}{G_{k}}\rangle -\frac{G_{y}^{in}}{h}\left[\overset{0}{w}\right](-a). \end{array}$$

Balance conditions on the power balance and unambiguity of displacements while moving around the thin defect must be added to the solution of the external problem:(10)$$\begin{array}{l} \int_{-a}^{a}f_{3}^{}\left(\xi \right)d\xi =\;2h\left(\sigma_{xz}^{in}(a)-\sigma_{xz}^{in}(-a)\right),\;\\ \int_{-a}^{a}f_{6}^{}\left(\xi \right)d\xi =\left[w\right](a)-\left[w\right](-a). \end{array}$$

Solving (9) and (10) using the methods in [29,38,41], it is easy to obtain a system of linear algebraic equations with unknown coefficients of the decomposition of the jump functions $f_{r}(x)$ into a series by orthogonal Jacobi or Chebyshev polynomials.

An important aspect of the study of the strength of such structures is the improvement of their strength criteria. In fracture mechanics, it is acceptable to use the stress intensity factor to describe the behavior of the SSS in the vicinity of the crack tip [42,45,61,62,63,66]. This is not sufficient for the case of a thin deformable inclusion. In [45], the authors obtained the two-term asymptotical expressions for the distribution of SSS in the vicinity of the thin inclusion tips using the introduced generalized stress intensity factors (GSIF):(11)$$K_{31}+iK_{32}=\underset{r\to 0\;(\theta =0,\pi )}{\lim}\sqrt{2\pi r}\left(\sigma_{yz}+i\sigma_{xz}\right).$$

Here, (r,θ) is a system of polar coordinates with the origin near the right or the left tip of the inclusion z=±rexp(iθ)±a.

Considering the well-known mathematical analogy [67], the obtained solutions to the antiplane problem can be regarded as solutions to the accordant heat conduction problem, if we take into account the correspondence of the values
$$\begin{array}{l} w∼T,\;\frac{\partial w}{\partial x}∼\frac{\partial T}{\partial x},\;\frac{\partial w}{\partial y}∼\frac{\partial T}{\partial y},\;q_{x}∼\sigma_{xz},\;q_{y}∼\sigma_{yz},\;\\ G_{x}∼\lambda_{x},\;G_{y}∼\lambda_{y},\;K_{31}∼k_{qy},\;K_{32}∼k_{qx}. \end{array}$$

The terms are as follows: T—temperature, $q_{x},\;q_{y}$—heat flows, $\lambda_{x},\;\lambda_{y}$—thermal conductivity coefficients, $k_{qy},\;k_{qx}$—heat flow intensity factors [40].

## 4. Numerical Results and Discussion

Since the main focus of this article is to investigate the effect of the functional gradient on the mechanical properties of the inclusion material, we will limit ourselves to one of the most representative variants of the structure loading: homogeneous longitudinal shear $\sigma_{yz}^{\infty }=\tau$ and $\sigma_{xzk}^{\infty }=\tau_{k}\;\left(k=1,2\right)$ at infinity. However, the calculations for loading by concentrated force factors or dislocations do not make any fundamental difference except for the necessity to consider the locality of their application [29,41].

The dependence $G_{x}(x),\;G_{y}(x)$ on coordinate x for mathematical modeling can be defined as an arbitrary function (linear, exponential, power, periodic [58], etc.), which adequately reflects the desired practical properties of the material. To illustrate the method, let us consider one of the illustrative variants of the functional gradient of the inclusion material—the piecewise linear one:(12)$$G_{x}(x)=G_{y}(x)=\{\begin{array}{c} \left(G_{01}-G_{0}\right)\frac{x}{a}+G_{01},\;x\in \left[-a,0\right];\\ \left(G_{02}-G_{01}\right)\frac{x}{a}+G_{01},\;x\in \left[0,\;a\right], \end{array}$$
where $G_{0},\;G_{01},\;G_{02}$—some given constants.

To significantly reduce the number of calculations without loss of generality, it is convenient to use the following dimensionless quantities, marked by symbol “~” (tilde) on top:$$\begin{array}{l} \;\tilde{x}=x/a,\;\tilde{h}=h/a,\;\tilde{y}=y/a,\\ {\tilde{G}}_{x}^{in}(\tilde{x})=G_{x}^{in}(x)/G_{gav},\;{\tilde{G}}_{y}^{in}(\tilde{x})=G_{y}^{in}(x)/G_{gav},\\ \;{\tilde{\tau }}_{k}=\tau_{k}/G_{gav},\;\tilde{\tau }=\tau /G_{gav},\;G_{gav}=\sqrt{G_{1}G_{2}},\\ {\tilde{G}}_{0}=G_{0}/G_{gav},\;{\tilde{G}}_{01}=G_{01}/G_{gav},{\tilde{G}}_{02}=G_{02}/G_{gav},\\ {\tilde{\sigma }}_{xz}^{}(\tilde{x})=\sigma_{xz}^{}(x)/G_{gav},\;{\tilde{\sigma }}_{yz}^{}(\tilde{x})=\sigma_{yz}^{}(x)/G_{gav}. \end{array}\;{\tilde{K}}_{31}^{}=\frac{K_{31}^{+}}{2\tilde{C}G_{gav}\sqrt{\pi a}},\;{\tilde{K}}_{32}^{}=\frac{K_{32}^{+}}{2p_{2}G_{gav}\sqrt{\pi a}},$$
where $K_{31}^{+}$ $K_{32}^{+}$ are the GSIFs near the tip x=+a of the inclusion.

Figure 2, Figure 3, Figure 4, Figure 5, Figure 6, Figure 7, Figure 8, Figure 9, Figure 10 and Figure 11 illustrate the dependence of the stress–strain behavior of the matrix in the inclusion vicinity on the variation in the parameters ${\tilde{G}}_{0},\;{\tilde{G}}_{01},\;{\tilde{G}}_{02}$, the values of which were chosen to reveal a qualitative picture of the FGM effect on the stress–strain parameters. It can be immediately concluded from Figure 2 and Figure 3 that under the load $\tilde{\tau }$, the dimensionless ${\tilde{K}}_{31}^{+}$ are expected to decrease with the increasing shear moduli of any part of the inclusion, while at ${\tilde{K}}_{32}^{+}$ they appear to increase with increasing load ${\tilde{\tau }}_{k}$.

The effect of changes in the moduli ${\tilde{G}}_{x}(x),\;{\tilde{G}}_{y}(x)$ on the stresses ${\tilde{\sigma }}_{yz}^{}$, ${\tilde{\sigma }}_{xz}^{}$ on the inclusion surface is more obvious if we choose a linear growth law for them along the inclusion axis (Figure 4, Figure 5 and Figure 6). The magnitude of the surface stresses increases significantly in the stiffer part of the inclusion. The larger the stiffness gradient, the more significant the increase.

The choice of the piecewise linear law of moduli ${\tilde{G}}_{x}(x),\;{\tilde{G}}_{y}(x)$ change in the Formulae (14) as ${\tilde{G}}_{01}={\tilde{G}}_{02}$ (variant 1) or ${\tilde{G}}_{0}={\tilde{G}}_{01}$ (variant 2) has a more contrasting effect on the surface stresses ${\tilde{\sigma }}_{yz}^{}$, ${\tilde{\sigma }}_{xz}^{}$, especially in the vicinity of the gradient breaking point x=0 (Figure 7 and Figure 8). Moreover, variant 2 of the functional dependence of the inclusion material moduli leads to partial unloading in the softer part of the inclusion near the breaking point x=0 (Figure 8).

Figure 9, Figure 10 and Figure 11 illustrate the changes in the stress field in the matrix in the inclusion vicinity under different variants of the law of functional change of the inclusion material moduli. The trends towards a decrease in the stress magnitudes in the vicinity of the stiffer parts of the inclusion are visible.

## 5. Conclusions

The proposed sufficiently simple and mathematically correct methodology made it possible for us to construct, for the first time, a mathematical model of a deformable thin linear interfacial inclusion with essentially inhomogeneous linear mechanical properties. Such a model can be used to simulate a thin inclusion from a functionally graded material and to solve the corresponding problems of defining the stress–strain field of the corresponding micro- or nanostructures by efficient analytical–numerical methods (the jump function method and its modifications), without the need to involve purely numerical approaches (in particular, FEM).

The calculations of the stress–strain field components for simple test cases of the functional dependence of the shear moduli of inclusion material have demonstrated the expected qualitative picture of their effect on the variation in the FGM parameters. In particular, (1) the stress magnitude increases significantly in the vicinity of the inclusion regions with increased stiffness; (2) the combination of the inclusion materials from parts with piecewise linear mechanical characteristics may lead to partial unloading of the inclusion and matrix in their softer part in the vicinity of the breaking point of the gradient dependence of the inclusion material parameters; (3) the contrast of the stress field changes of the inclusion and matrix is proportional to the increase in the gradient dependence.

The conclusions and calculation results are easily transferable to analogous problems of thermal conductivity and thermoelasticity with possible frictional heat generation and can be used for recommendations on the optimal operating parameters of structures.

The discussed conclusions can be useful in designing the functionally gradient mechanical properties of the material of inclusions and in the optimization of engineering structures to increase their strength and service life. The proposed method is effective for solving a wide class of problems of deformation of solids with thin deformable inclusions of finite length and can be used for SSS calculation for different FGM inclusions.

## Figures and Tables

**Figure 1 materials-15-08591-f001:**
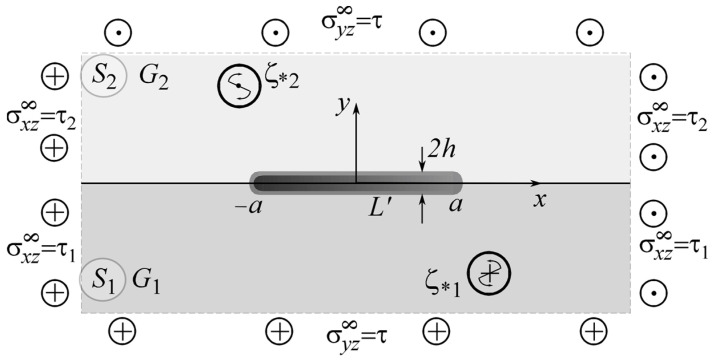
The loading and geometric scheme of the problem.

**Figure 2 materials-15-08591-f002:**
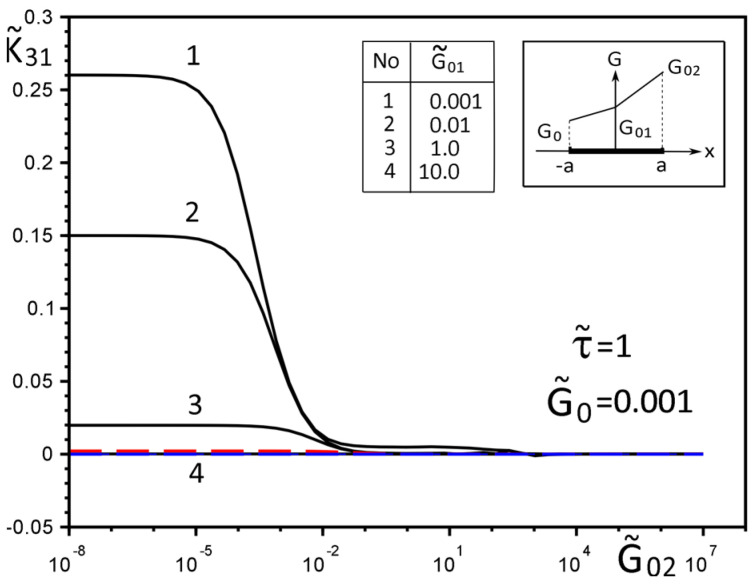
Influence of the parameters ${\tilde{G}}_{0},\;{\tilde{G}}_{01},\;{\tilde{G}}_{02}$ on the GSIF ${\tilde{K}}_{31}^{+}$ under the load, uniformly distributed on infinity stress $\sigma_{yz}^{\infty }=\tau$.

**Figure 3 materials-15-08591-f003:**
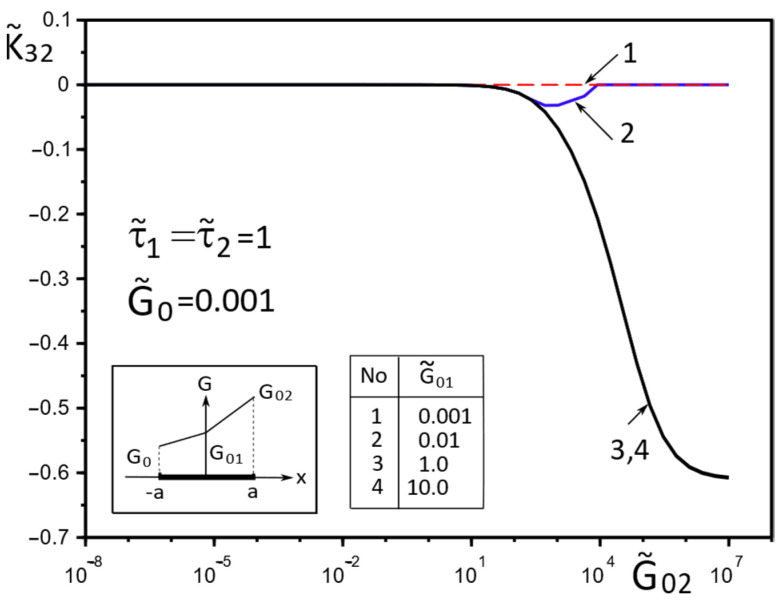
Influence of the parameters ${\tilde{G}}_{0},\;{\tilde{G}}_{01},\;{\tilde{G}}_{02}$ on the GSIF ${\tilde{K}}_{32}^{+}$ under the load, uniformly distributed on infinity stress $\sigma_{xzk}^{\infty }=\tau_{k}\;(k=1,2)$.

**Figure 4 materials-15-08591-f004:**
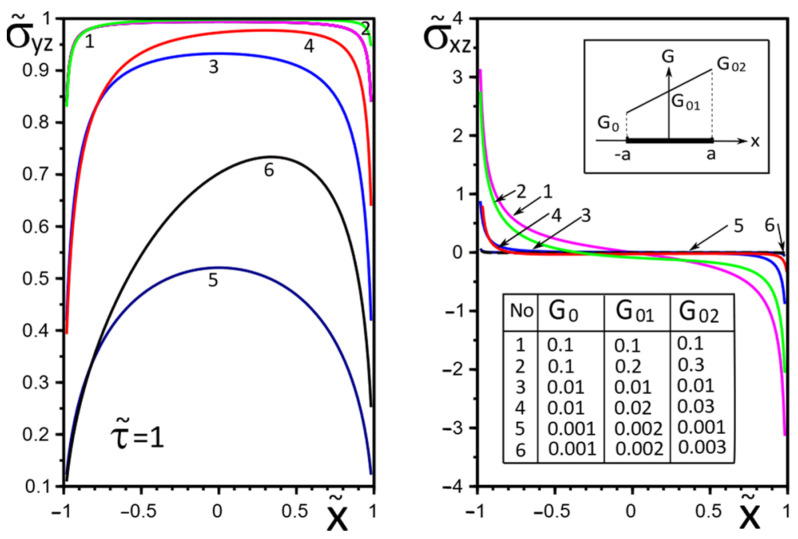
Stress distribution along with the upper interface (inclusion–matrix half-space $S_{2}$) with a linear distribution of material stiffness.

**Figure 5 materials-15-08591-f005:**
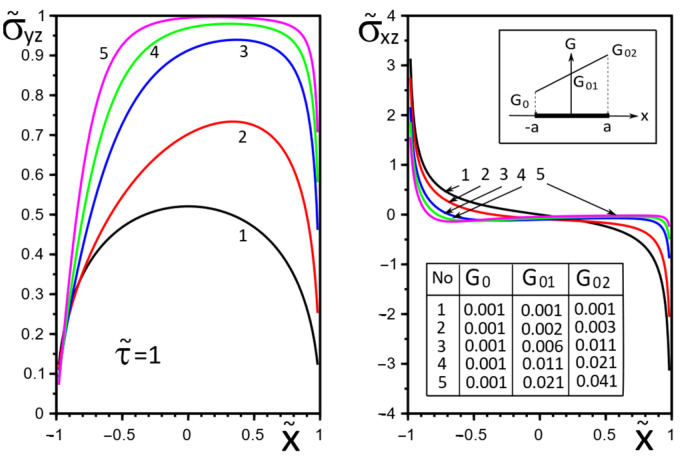
Stress distribution along with the upper interface (inclusion–matrix half-space $S_{2}$) with a linear distribution of material stiffness.

**Figure 6 materials-15-08591-f006:**
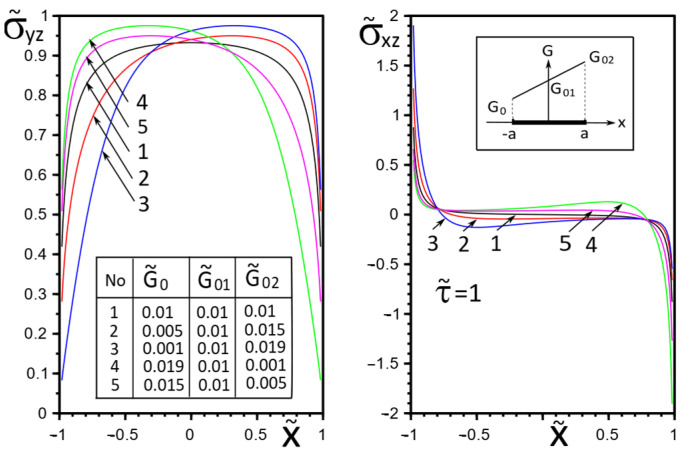
Stress distribution along with the upper interface (inclusion–matrix half-space $S_{2}$) with a linear distribution of material stiffness.

**Figure 7 materials-15-08591-f007:**
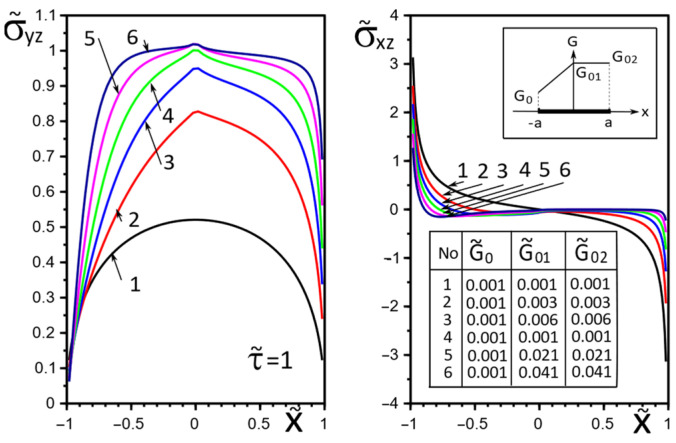
Stress distribution along with the upper interface (inclusion—matrix half-space $S_{2}$) with a piecewise linear distribution of material stiffness.

**Figure 8 materials-15-08591-f008:**
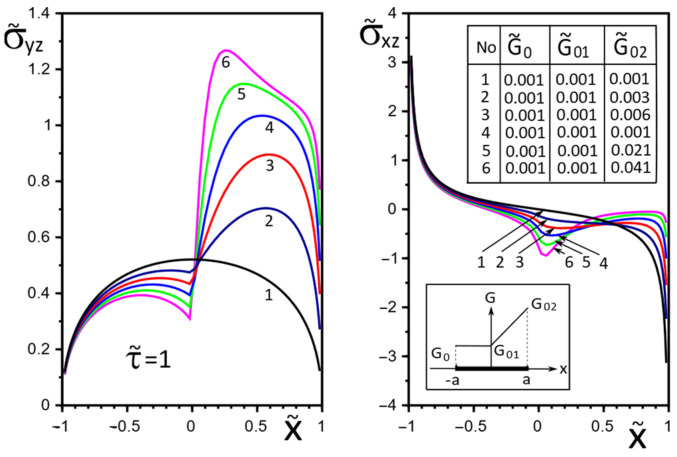
Stress distribution along with the upper interface (inclusion—matrix half-space $S_{2}$) with a piecewise linear distribution of material stiffness.

**Figure 9 materials-15-08591-f009:**
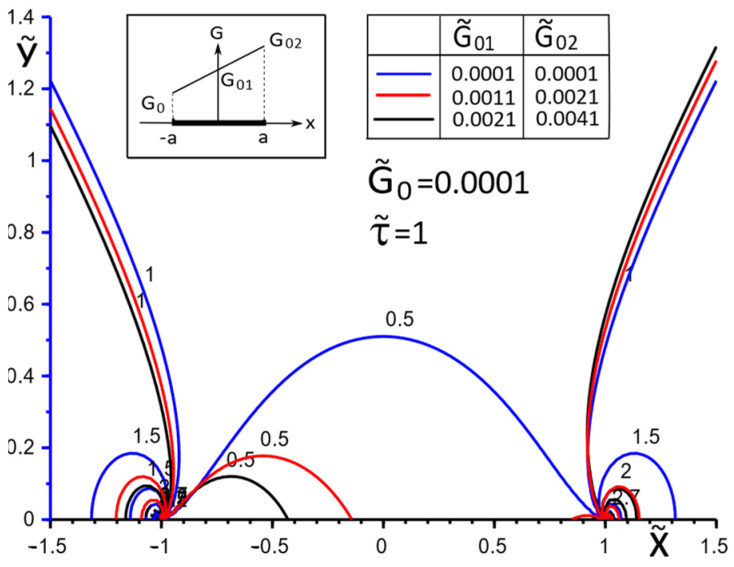
Stress distribution in the matrix at the vicinity of the inclusion with a linear distribution of material stiffness.

**Figure 10 materials-15-08591-f010:**
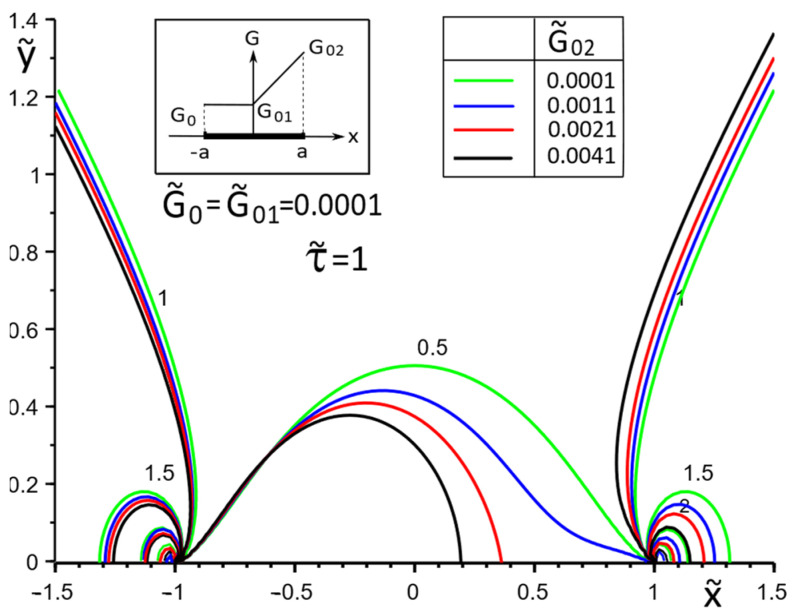
Stress distribution in the matrix at the vicinity of the inclusion with the piecewise linear distribution of material stiffness.

**Figure 11 materials-15-08591-f011:**
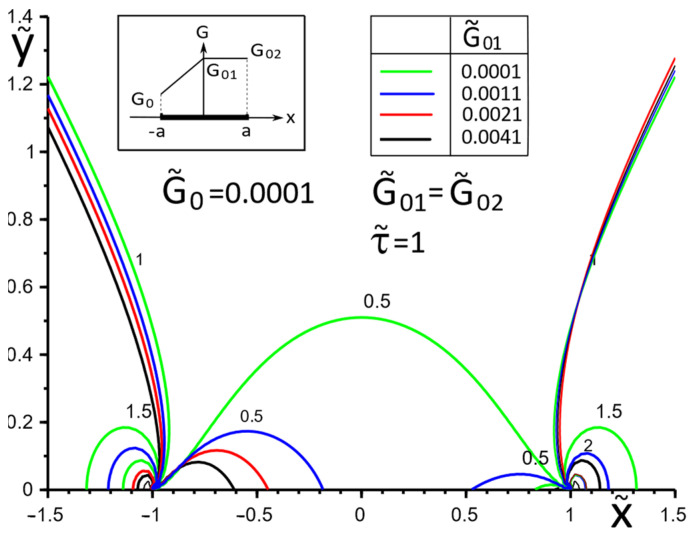
Stress distribution in the matrix at the vicinity of the inclusion with the piecewise linear distribution of material stiffness.

## Data Availability

The data presented in this study are openly available at https://ua1lib.org/book/665574/5c937e (accessed on 9 January 2022), reference number [37]; https://doi.org/10.3390/ma15041435 (accessed on 9 January 2022), reference number [38]; and https://doi.org/10.3390/ma14174928 (accessed on 9 January 2022), reference number [41].

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
