# Peer review of "Effect of the Transverse Functional Gradient of the Thin Interfacial Inclusion Material on the Stress Distribution of the Bimaterial under Longitudinal Shear"

_materials, 2022, doi:10.3390/ma15238591_

Round 1
Reviewer 1 Report
1. “a mathematical model of such an FGM inclusion is constructed”, “An” should be “a”.
2. “Contact between the bimaterial structure components between matrix components and inclusion at…”, it is recommended to modify the syntax here to avoid ambiguity.
3.“it is convenient to use the following dimensionless quantities:”,the symbol of dimensionless parameter “G0,G01,G02” need to be modified so that it is not the same as the original parameter symbol.
4. For the curves in the paper, it is recommended to unify the style, such as using all color curves.
5. For the discussion of the results, the curve change trend is simply described, which is too general. It is suggested to give a more in-depth analysis in combination with the theoretical formula.
6. In the title, “on the Strength Parameters”, it is suggested to change it to “on the Stress Distribution”, which should be more appropriate.
Author Response
We are grateful for an objective review , which helped us significantly improve the article. The title is changed to “on the Stress Distribution”,
Our best regards, Liubov and Yosyf Piskozub, Heorhiy Sulym

Reviewer 2 Report
Good work, I see you missed some key points. Mathematical modeling of nanostructures requires the construction of more complex constitutional laws in comparison with the macro level
Figure 4-5 how did you check them experimentally?
Figure 10: the effect is similar to inclusions in FSW
State of the art: frictional processes are key for MDPI, Joining metrics enhancement when combining FSW and ball-burnishing in a 2050 aluminum alloy, Surface and Coatings Technology 367, 327-335 and Burnishing of FSW aluminum Al–cu–li components, Metals 9 (2), 260 Please there are a few works related, the two ones are basics.
Maths seems Ok, please rewrite introduction and discussion.
Author Response
We are grateful for an objective review , which helped us significantly improve the article.
Our best regards, Liubov and Yosyf Piskozub, Heorhiy Sulym

Reviewer 3 Report
A simple mathematical model is proposed for describing stress-strain relation of the deformable thin linear interfacial inclusion with inhomogeneous linear mechanical properties.
Remarks:
1. Currently gradient of the inclusion material is modelled as piecewise linear – the simplest one. Please discuss in introduction or in chapter 4 also some other widely used gradient functions like exponential, power law, four parameter power low. Four parameter trigonometric, etc. (see doi:10.1088/1757-899X/1140/1/012013)
in regard to
a) possible impact on complexity of solution (does it remains in the same range)
b) possible impact on results
c) is it foreseen to extend results for other grading functions?
2. More precise description of the numerical method/approach used should be provided current version „Solving the (9)-(10) using the methods [29, 38, 41]“ does not say actually anything about methods used. Another recent and accurate approaches can be also discussed: like HOHWM (DOI 10.1007/s11029-021-09929-2) , GDQM (DOI:10.1142/9781860949524_0150)
Author Response

(The authors gave the same response as above.)

Round 2
Reviewer 2 Report
Ok